# Evaluation of Load on Cervical Disc Prosthesis by Imposing Complex Motion: Multiplanar Motion and Combined Rotational–Translational Motion

**DOI:** 10.3390/bioengineering11080857

**Published:** 2024-08-22

**Authors:** Hossein Ansaripour, Stephen J. Ferguson, Markus Flohr

**Affiliations:** 1Institute for Biomechanics, D-HEST, ETH Zurich, Gloriastrasse 37/39, 8092 Zurich, Switzerland; sferguson@ethz.ch; 2CeramTec GmbH, CeramTec-Platz 1-9, 73207 Plochingen, Germany; m.flohr@ceramtec.de

**Keywords:** in vitro kinematics, cervical disc prosthesis, multiplanar motion, subluxation

## Abstract

(1) Background: The kinematic characteristics of disc prosthesis undergoing complex motion are not well understood. Therefore, examining complex motion may provide an improved understanding of the post-operative behavior of spinal implants. (2) Methods: The aim of this study was to develop kinematic tests that simulate multiplanar motion and combined rotational–translational motion in a disc prosthesis. In this context, five generic zirconia-toughened alumina (BIOLOX^®^*delta*, CeramTec, Germany) ball and socket samples were tested in a 6 DOF spine simulator under displacement control with an axial compressive force of 100 N in five motion modes: (1) flexion–extension (FE = ± 7.5°), (2) lateral bending (LB = ± 6°), (3) combined FE-LB (4) combined FE and anteroposterior translation (AP = 3 mm), and (5) combined LB and lateral motion (3 mm). For combined rotational–translational motion, two scenarios were analyzed: excessive translational movement after sample rotation (scenario 1) and excessive translational movement during rotation (scenario 2). (3) Results: For combined FE-LB, the resultant forces and moments were higher compared to the unidirectional motion modes. For combined rotational–translational motion (scenario 1), subluxation occurred at FE = 7.5° with an incremental increase in AP translation = 1.49 ± 0.18 mm, and LB = 6° with an incremental increase of lateral translation = 2.22 ± 0.16 mm. At the subluxation point, the incremental increase in AP force and lateral force were 30.4 ± 3.14 N and 40.8 ± 2.56 N in FE and LB, respectively, compared to the forces at the same angles during unidirectional motion. For scenario 2, subluxation occurred at FE = 4.93° with an incremental increase in AP translation = 1.75 mm, and LB = 4.52° with an incremental increase in lateral translation = 1.99 mm. At the subluxation point, the incremental increase in AP force and lateral force were 39.17 N and 38.94 N in FE and LB, respectively, compared to the forces in the same angles during the unidirectional motion. (4) Conclusions: The new test protocols improved the understanding of in vivo-like behavior from in vitro testing. Simultaneous translation–rotation motion was shown to provoke subluxation at lower motion extents. Following further validation of the proposed complex motion testing, these new methods can be applied future development and characterization of spinal motion-preserving implants.

## 1. Introduction

Anterior cervical discectomy and fusion (ACDF) is the gold standard for treatment of degenerative disc disease causing myelopathy or radiculopathy [1]. There are outstanding short- and long-term clinical results for ACDF [2,3,4,5]. However, Baba et al. found the development of new dynamic spinal canal stenosis in 25% of patients after a mean of 8.5 years of follow-up [6]. In another study, an adjacent segment disease (ASD) rate of 2.9% per year was reported after fusion [7]. Several in vitro biomechanical tests have also proposed that the considerable increase in range of motion (ROM) and intradiscal pressure (IDP) at the adjacent segments may contribute to the development of ASD following ACDF [8]. Concern about adjacent segment disease has prompted a surge in the development of motion-preserving disc prostheses.

Disc prostheses has been developed to preserve motion across the spine, alter IDP less at the adjacent segments, provide pain relief, and prevent the development of late ASD [8]. A 5-year meta-analysis revealed a relatively low rate of complications for cervical disc prostheses (0–4.0%) and lumbar disc prostheses (0–16.7%) [9]. Nevertheless, disc arthroplasty is a relatively new treatment, and long-term data on issues including fatigue failure, wear debris accumulation, implant luxation, and heterotopic ossification are insufficient [10,11,12,13]. Moreover, a cross-sectional analysis reported complications mainly related to implant migration, insertion problems, neck pain, heterotopic ossification, and radiculopathy following cervical disc arthroplasty [14]. These issues might be traced back to design, material selection, or surgical error [15].

Biomechanical in vitro testing is useful for the preclinical assessment of spinal implant performance and the success of operative procedures. Nonetheless, the majority of research has focused on observing simple arcs of motion in a single plane (i.e., flexion-extension (FE), lateral bending (LB), and axial rotation (AR)), making it challenging to address clinical issues such as implant luxation and migration [8]. A previous study examined cervical spine kinematics in multiplanar motion after disc replacement and ACDF [16]. They hypothesized that examining multiplanar motion could provide a better understanding of the in vivo behavior of spinal implants [16]. Penning et al. also suggested that the segmental motion of the cervical spine was not simply a rocking movement but accompanied by a displacement between vertebrae [17]. Hence, the absence of translational motion in in vitro testing can prevent the complete understanding of disc prosthesis function. For instance, abnormal or excessive vertebral translations may serve as clinically important indicators for the evaluation of disc subluxation and dislocation risk.

This study aimed to develop kinematic test methods for a cervical disc prosthesis that simulate multiplanar motion and combined rotational–translational motion. The specific research objectives were as follows: (1)Quantifying the forces and moments generated during the multiplanar motion (combined FE-LB). In this case, three component forces (i.e., Fx, Fy, and Fz) and three component moments (i.e., Tx, Ty, and Tz) were calculated, and the resultant values for forces and moments were then compared between the unidirectional motion test and the multiplanar motion test.(2)Determining the conditions under which subluxation or dislocation occurs during combined rotational–translational movements. In this case, the lateral force, the degree of the rotation, and the translation were measured until the samples reached the subluxation point.

By focusing on these specific metrics, this research provides detailed insights into the in vitro behavior of cervical disc prostheses under complex loading conditions.

## 2. Materials and Methods

### 2.1. Experimental Components

Five generic zirconia-toughened alumina (BIOLOX^®^*delta*, CeramTec, Plochingen, Germany) ball and socket samples (shape: cylindrical body; height: 20 mm; diameter: 11 mm) were tested in a 6 DOF spine simulator (material testing system MTS-370.02 Bionix, Eden Prairie, MN USA) (Figure 1). The rotations in FE and LB are achieved by a rotary gimbal drive around the y and x axis, respectively (Figure 1). A 6-component load cell (FT 15954, mini-45/SI-580–20) was attached between the upper fixture and the gimbal for data acquisition at a 100 Hz sampling frequency to record reaction moments and forces. Alignment of the sample and setup components was required during the test run to assure unconstrained rotation around the desired axis and to avoid excessive forces and moments and, as a result, damage to the sample. The X/Y table can provide appropriate translational compensation, which corrects for the offset between the testing machine’s center of rotation (COR) and the sample’s COR. In this study, the offset measurement was 95 mm. Therefore, a translational compensation of 12.40 mm for FE (7.5°) and 9.93 mm for LB (6°) were required (Offset×sinα). The orientation of the coordinate system was the same as that suggested in the literature [18].

### 2.2. Study Design

The fixtures for the specimens were attached to the gimbal (upper part) and the X/Y table (lower part) of the test rig (Figure 1). The samples were tested under displacement control with a constant axial compressive force of 100 N in five motion modes: (1) FE (±7.5°), (2) LB (±6°), (3) combined FE-LB (Figure 2a), (4) combined FE and AP (3 mm), and (5) combined LB and lateral motion (3 mm). Two scenarios were studied for the combined rotational–translational motion:Scenario 1: The application of an excessive translational movement (3 mm) subsequent to the complete rotation of the sample (Figure 2b). Before applying the excessive translation, the samples remained in a fully flexed condition for 10 seconds to reach the required values for rotation and axial force (i.e., the dwelling phase). During the rotation step, the applied translation was due to the offset between the testing machine’s COR and the sample’s COR (12.40 mm for FE and 9.93 mm for LB) in order to adjust the alignment.Scenario 2: The concurrent application of excessive translational motion (3 mm) and rotation (Figure 2c). A translational adjustment (command normal translation in Figure 2c) was made during the rotation to prevent excessive forces and moments due to the offset between the sample’s COR and the testing machine’s COR. The 3 mm of extra translation was then superimposed on the translational adjustment.

A ramp load profile and a sinusoidal load profile were applied for scenario 1 and scenario 2, respectively (Figure 2b,c) to assess its potential impact on results. It is noteworthy that both ramp and sinusoidal load profiles are prevalent in the literature and exhibit negligible difference between them [8]. Nevertheless, the sinusoidal load profile bears closer resemblance to physiological conditions.

The objective of these excessive translations was to illustrate the response of the disc prosthesis under severe loading conditions.

For FE, LB, and combined FE-LB, the tests were conducted over five cycles, with the first four cycles serving as preconditioning cycles and not being included in later analysis. For multiplanar motion and unidirectional motion tests, an X/Y table with passive linear bearings (Figure 3) was employed, allowing for the sample to move freely while rotating.

For the rotational–translational motion test, a hydraulically controlled X/Y table (Figure 4) was employed to apply the required lateral movement. In earlier laboratory procedures, tests were undertaken to optimize setup design by decreasing setup friction’s effect on the outcomes. Furthermore, the interface of the samples was lubricated with Ringer’s solution (i.e., an isotonic solution analogous to an animal’s body fluids) to reduce the friction between articulating surfaces.

**Figure 2 bioengineering-11-00857-f002:**
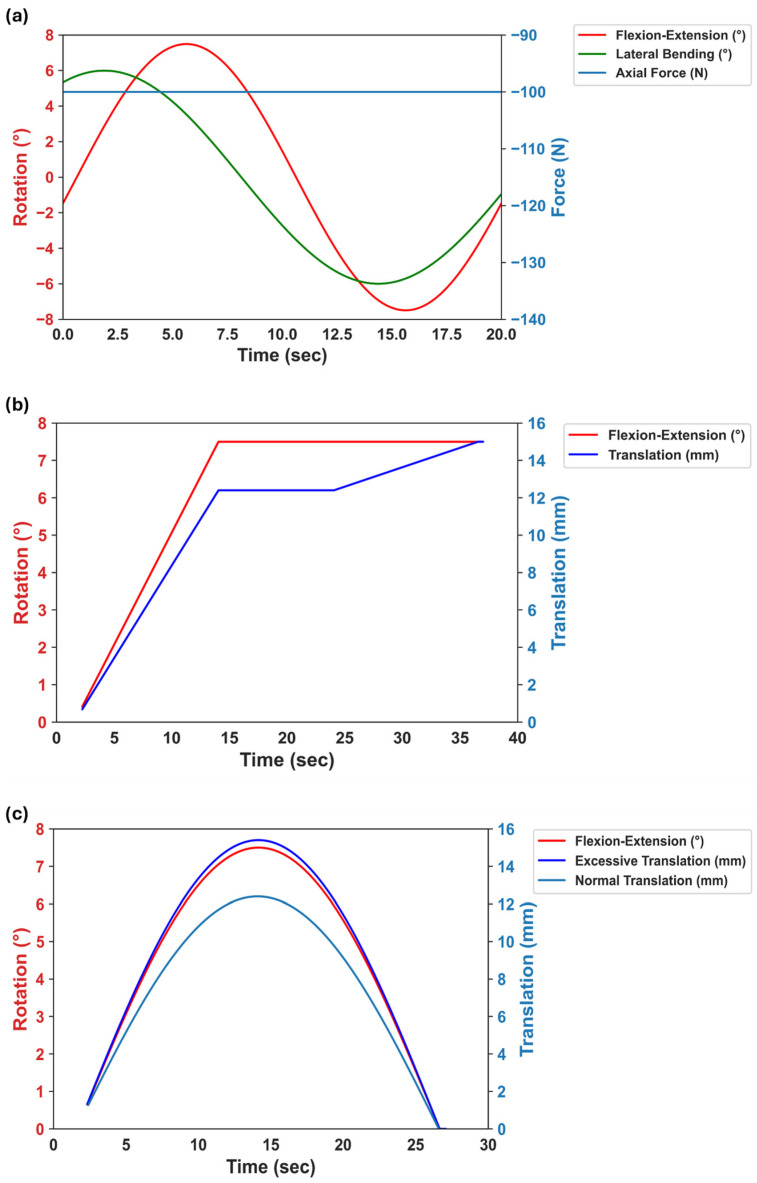
Illustration of load profiles: (**a**) Multiplanar motion profile (combined FE-LB) with 100 N of axial compressive force during the test. (**b**) Excessive translation after flexion. (**c**) The application of the excessive translation while the sample flexed. During rotational–translational motion experiments, a constant 100 N axial compressive force was also applied.

**Figure 3 bioengineering-11-00857-f003:**
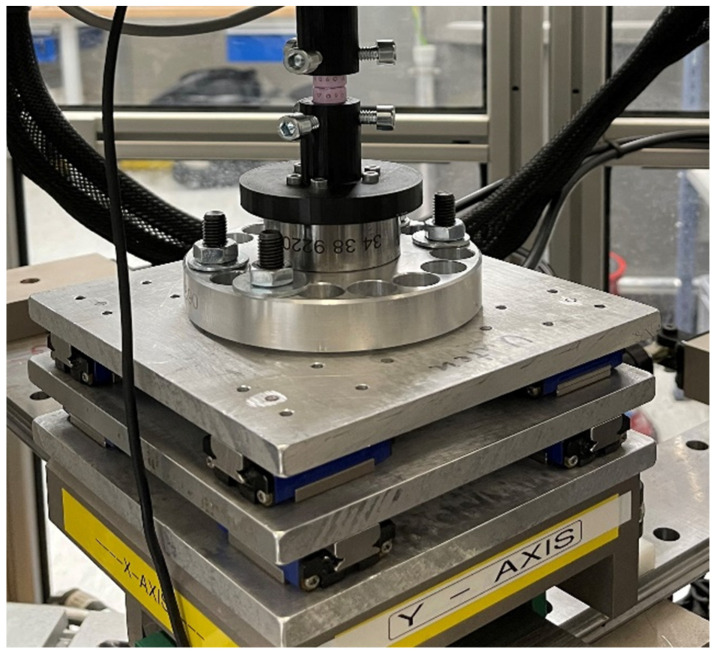
Passive linear bearing X/Y table.

**Figure 4 bioengineering-11-00857-f004:**
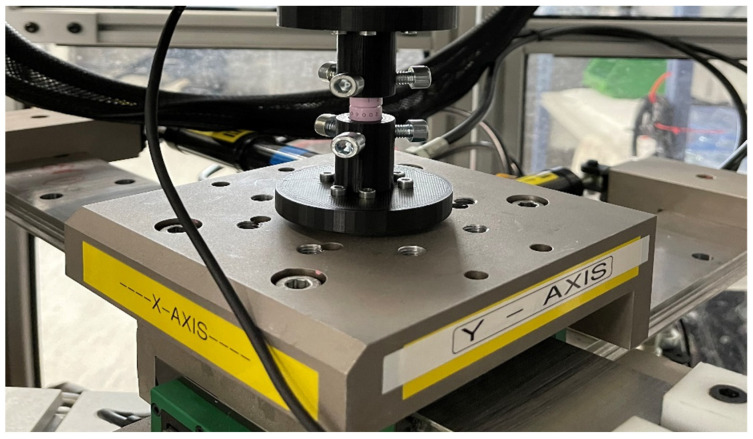
Hydraulic X/Y table.

### 2.3. Postprocessing

No data transformation was required since the rotation center of the sample was aligned with the vertical axis of the load cell during the test. The load cell data were filtered using a Savitzky–Golay filter to smooth the data without distorting the signal tendency. Customized Python code (Python 3.8.5 programming language) was used for data analysis to derive the moments and forces of the system.

#### 2.3.1. Unidirectional and Multiplanar Motion

In the absence of facet joints and ligaments, the reaction shear forces and moments are primarily a function of disc friction, setup friction, and the applied axial force (both magnitude and orientation). In other words, the combination of angular motion and eccentric axial force results in anteroposterior (AP) and mediolateral (ML) forces and moments in FE and LB, respectively (Figure 5). For each test, 3 component forces (i.e., Fx, Fy, and Fz) and 3 component moments (i.e., Tx, Ty, and Tz) of the system were calculated. The resultant values for forces and moments were then compared between the unidirectional motion test and the multiplanar motion test.

#### 2.3.2. Rotational–Translational Motion

For the rotational–translational motion experiments, the lateral force and translation were measured until the samples reached the subluxation point, followed by micro-separation.

## 3. Results

### 3.1. Reaction Moments and Forces for Combined and Unidirectional Motion Tests

In coupled motion consisting of FE-LB, the absolute mean of the maximum (std) Tx (LB moment), Ty (FE moment), and Tz (axial torque) were 0.56 (0.01) Nm, 0.82 (0.1) Nm, and 0.01 (0.003) Nm, respectively (Figure 6a and Table 1). For the simple arcs of FE, the absolute mean of the maximum (std) Tx, Ty, and Tz were 0.15 (0.06) Nm, 0.74 (0.12) Nm, and 0.01 (0.01) Nm, respectively (Figure 6a and Table 1). For the simple arcs of LB, the absolute mean of the maximum (std) Tx, Ty, and Tz were 0.67 (0.08) Nm, 0.1 (0.07) Nm, and 0.01 (0.003) Nm, respectively (Figure 6a and Table 1).

For combined FE-LB, the absolute mean of the maximum (std) Fx (anteroposterior force), Fy (lateral force), and Fz (axial force) were 20.92 (1.84) N, 13.40 (0.60) N, and 106.61 (13.61) N, respectively (Figure 6b and Table 1). For the simple arcs of FE, the absolute mean of the maximum (std) Fx, Fy, and Fz were 17.15 (2.14) N, 3.86 (1.09) N, and 103.38 (29.94) N, respectively (Figure 6b and Table 1). For the simple arcs of LB, the absolute mean of the maximum (std) Fx, Fy, and Fz were 2.57 (1.004) N, 14.87 (1.65) N, and 102.46 (24.33) N, respectively (Figure 6b and Table 1).

Table 2 indicates that the combined FE-LB has a greater resultant moment and force compared to FE and LB individually. The resultant moment and force obtained from the FE and LB are similar.

### 3.2. Shear Forces, Translations, and Degree of Rotations for Rotational–Translational Motion Tests

At 12.4 mm and 9.9 mm, the mean (std) shear forces in disc-related axes for FE and LB, respectively, were 47.4 (6.40) N and 40.1 (4.27) N (Figure 7). Then, the excessive lateral translation was applied to reach the micro-separation or subluxation point. After complete flexion, the samples reached the subluxation point at an absolute mean (std) displacement of 13.8 (0.2) mm and absolute mean (std) shear force of 77.0 (7.1) N (Figure 7a). Thus, incremental increases in AP translation = 1.49 ± 0.18 mm and shear force = 30.4 ± 3.14 N were observed at subluxation point. After completing lateral bending rotation, the samples reached the subluxation point at an absolute mean (std) displacement of 12.1 (0.2) mm and absolute mean (std) shear force of 80.9 (6.3) N (Figure 7b). Thus, an incremental increase in lateral translation = 2.22 ± 0.16 mm and shear force = 40.8 ± 2.56 N were observed at the subluxation point.

Due to the complex physiological condition, we hypothesized that the subluxation may also occur during rotation. Thus, the excessive lateral translation was superimposed on the translational compensation while the sample was rotating. During the flexion, the sample reached the subluxation point at FE = 4.9°, lateral displacement = 9.9 mm, and shear force = 64.9 N (Figure 8). During LB, the sample reached the subluxation point at LB = 4.5°, lateral displacement = 9.5 mm, and shear force = 72.8 N (Figure 9). For FE = 4.9°, lateral displacement and shear force were 8.1 mm and 25.7 N, respectively, under normal condition (i.e., motion to a predefined physiological angle, without additional translation). For LB = 4.5°, lateral displacement and shear force were 7.5 mm and 33.8 N, respectively, under normal condition. Therefore, an incremental increase in AP translation = 1.75 mm and lateral translation = 1.99 mm was observed at the subluxation point for FE and LB, respectively. Moreover, the incremental increase in AP shear force and lateral shear force was 39.17 N and 38.94 N for FE and LB, respectively, at the subluxation point.

## 4. Discussion

The first objective of this work was to compare the loads and moments in multiplanar motion tests (i.e., FE-LB) to those obtained in unidirectional motion tests (i.e., FE and LB). The results revealed that the resultant loads and moments for multiplanar motion were greater than those obtained for unidirectional testing. Since the samples were generic ball and socket components with no design constraints, one would expect that multiplanar motion test findings should be comparable to unidirectional motion test results. The slight discrepancy could be attributed to the fact that the friction regime is different in multiplanar motion test. It is recommended that the test be conducted on different designs (such as non-symmetrical designs in different planes) and non-articulating disc prostheses. This would comprehensively evaluate disc prosthesis response to complex motion, which is likely to occur in vivo.

In a previous study, cervical spine kinematics were examined in multiplanar motion after disc replacement utilizing human cadaver spines [16]. They found that the stiffnesses in multiplanar motion were higher than those in unidirectional motion. Due to the use of cadaver specimens in prior studies, our findings were not directly comparable, however consistent. The present study also provides insight into the influence of the testing protocol on common laboratory simulator results. Furthermore, the length of the evaluated spinal segment could affect the moment arm, resulting in greater moments for the same range of motion (ROM). In this study, rotation combined with a constant axial force was directly applied to the disc prosthesis, whereas in the literature, a pure bending moment was applied at the cranial level of C4–7 [16]. It would be beneficial to simulate facet joint loads and ligament contributions in the study. One approach is to use synthetic specimens that replicate cadaveric biomechanics and provide advantages for cross-laboratory validation studies while lowering costs and disease transmission [19,20].

This study also investigated subluxation, which is one of the most important clinical concerns following disc replacement. There have been two reports of disc dislocation in the literature [21]. The first report indicated that the prosthesis was implanted too anteriorly. The second was a technical error during implantation when the polyethylene inlay was not completely snapped into the inferior endplate. The improper snapping of the inlay into the metallic endplate, followed by the presence of shear forces, increases the likelihood of subluxation and dislocation. Furthermore, the incision of ligaments during surgery or muscle dysfunction can lead to hypermobility of the spine and an increase in shear loads on disc prosthesis, which increases the probability of migration. It is important to note that unstable fusion at the bone–implant interface may also contribute to disc dislocation. In this study, kinematic tests were prescribed to replicate subluxation by applying excessive translation during and after rotation. In both instances, the shear load increased continuously until the subluxation point. Furthermore, the tests revealed an increased risk of subluxation for complex, combined motion than for sequential movements.

In normal conditions (i.e., motion to a predefined physiological angle, without additional translation), the results indicated a linear relationship between the increase in shear loads and the rotation angle. However, the presence of excessive translation after full rotation caused a sharp increase in shear loads because the ball and socket samples had three degrees of freedom, permitting 3D rotations without translation (Figure 7). The same trend was also observed when excessive translation was applied during rotation (Figure 8 and Figure 9). It can be inferred that mobile core prostheses can provide a more effective resistance to aberrant situations. In addition, previous research demonstrated other advantages of mobile core prostheses, in that they can disperse pressure at the bone–prosthesis interface due to the capability of anteroposterior translation as well as having less variation in the instantaneous axis of rotation [22,23]. However, the long-term clinical outcome of mobile-core prostheses is still debated.

In subluxation testing, the value of shear loads and excess translation at the micro-separation point for LB were somewhat greater than those for FE. Furthermore, in the event of simultaneous rotational and translational motion, the subluxation occurred at comparable degrees of rotation for FE and LB because the ball and socket contact point did not shift during the kinematic tests. To determine whether there are significant differences between FE and LB for other devices, further studies are required to develop subluxation tests for non-symmetrical and non-articulating designs.

It is currently challenging to pinpoint what shear force magnitude leads to subluxation of discs in an in vivo condition. Based on the present results, even a modest increase in shear loads is enough to cause subluxation in this design. The development of subluxation tests using spinal loading simulators (with or without cadaver specimens) and finite element simulations in the future is crucial for a better understanding multi-directional loading conditions and their consequences.

The wear analysis of certain materials, such as ceramic, is difficult due to their remarkable resistance to low friction under lubricated conditions. Thus, the development of severe loading conditions such as subluxation or edge loading could be beneficial for a tribology investigation of the materials for TDR applications. At present, there is an absence of prior research that investigates the effects of edge loading on disc prosthesis deterioration. Notwithstanding this, a number of research studies examined the effects of edge loading on hip replacements, which may shed light on analogous phenomena affecting spinal disc prostheses. In metal-on-metal articulations for total hip replacements, edge loading resulted in pseudotumor formation, metallosis, aseptic lymphocytic vasculitis-associated lesions, and accelerated wear of the entire articulation [24,25,26,27,28]. In ceramic-on-ceramic articulations for total hip replacements, edge loading led to squeaking, stripe wear, and accelerated wear of the whole joint [29,30,31,32].

## 5. Conclusions

This study developed new test protocols for determining kinetic parameters of disc prosthesis during multiplanar motion (i.e., FE-LB) and subluxation tests. For the multiplanar motion test, the resultant forces and moments were slightly higher but comparable to the unidirectional motion test (i.e., FE and LB). For the subluxation test (scenario 1), the application of excessive translation led to a 62.44% and 101.74% increase in shear forces for FE and LB, respectively. For the subluxation test (scenario 2), the application of excessive translation resulted in a 152.53% and 115.38% increase in shear forces for FE and LB, respectively. The highest shear force was found at the subluxation point, followed by a sharp reduction due to complete dislocation. Multiplanar motion and subluxation testing will help develop and evaluate disc prosthesis function, longevity, and safety, as well as address clinical issues.

## Figures and Tables

**Figure 1 bioengineering-11-00857-f001:**
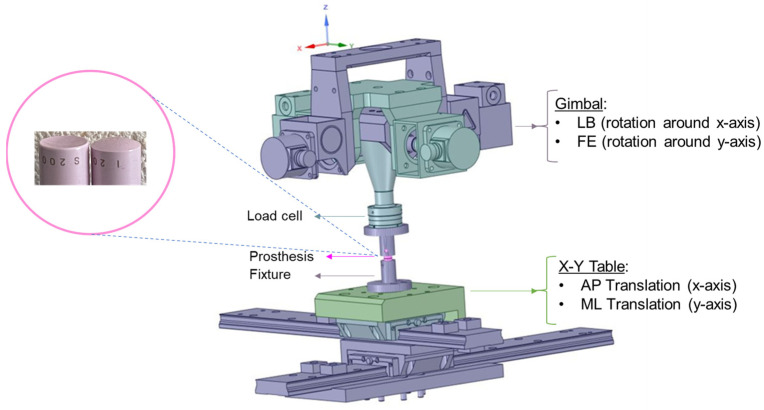
The spine simulator is equipped with a gimbal drive (providing FE and LB rotations) and an X/Y table (providing unconstrained translational motions). Zirconia-toughened alumina samples (pink ball and socket) were tested in a material testing machine.

**Figure 5 bioengineering-11-00857-f005:**
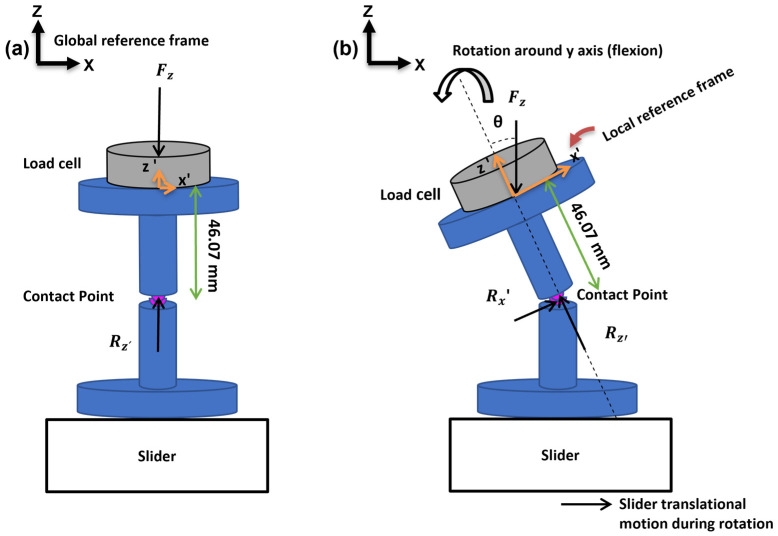
(**a**) A pure axial force is present in the neutral position and (**b**) the prosthesis flexes around the y-axis and the axial force becomes eccentric. Therefore, it results in lateral force and moments. The distance between the load cell’s origin and the sample’s contact point is 46.07 mm. R_x’_ and R_z’_ show the shear force and axial force, respectively, in the local reference frame (i.e., the implanted related axis). F_z_ shows the axial force in the global reference frame.

**Figure 6 bioengineering-11-00857-f006:**
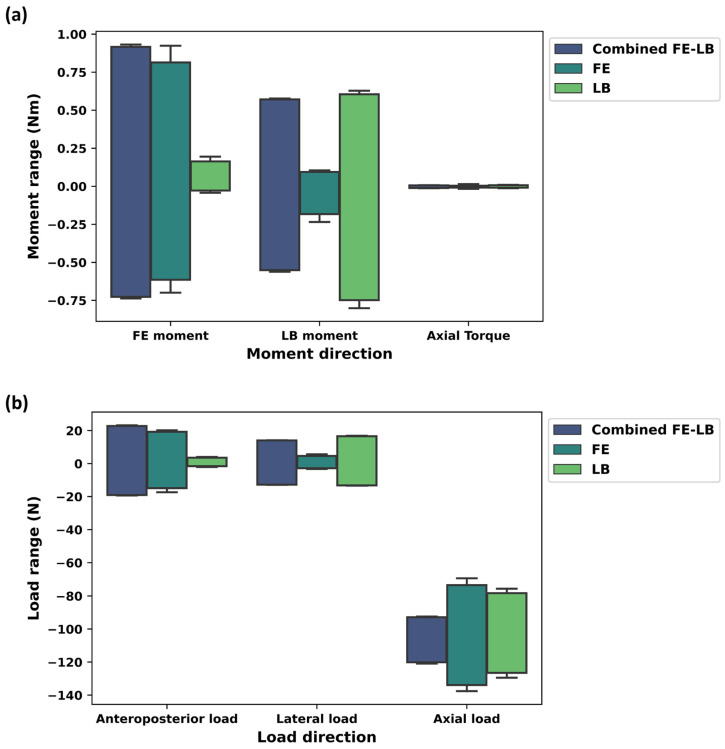
(**a**) Moment range (Nm) and (**b**) force range (N) for combined FE-LB, FE, and LB. In this graph, FE moment, LB moment, and axial torque represent Ty, Tx, and Tz, respectively. Anteroposterior force, lateral force, and axial force represent Fx, Fy, and Fz, respectively.

**Figure 7 bioengineering-11-00857-f007:**
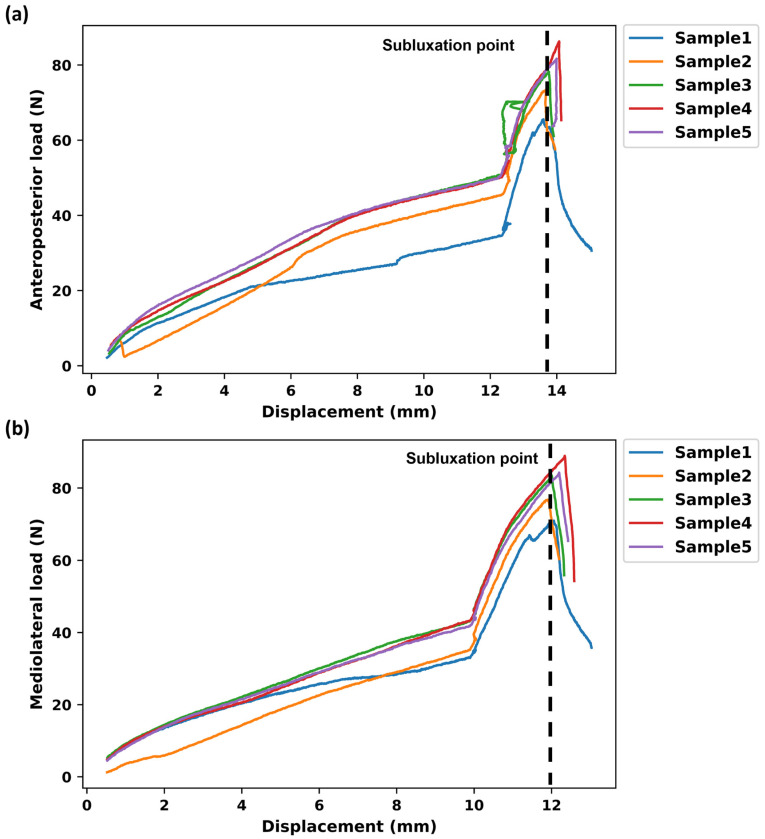
Subluxation test after (**a**) complete flexion and (**b**) complete lateral bending. Anteroposterior load and mediolateral load represent shear loads in the implanted related axes. The excessive translation of 3 mm was applied after the sample was fully rotated. The shear load and excessed translation were determined at the subluxation point or micro-separation site.

**Figure 8 bioengineering-11-00857-f008:**
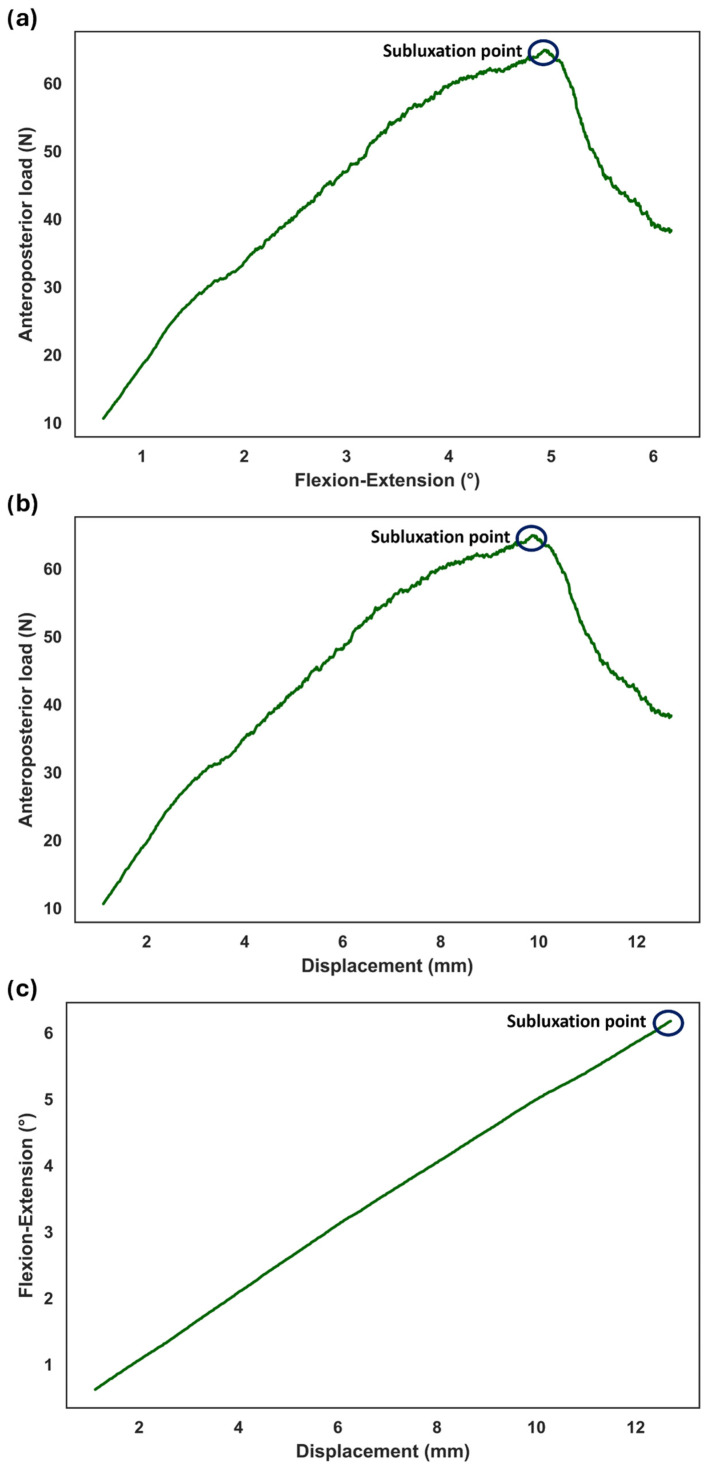
Subluxation test during flexion. Anteroposterior load represents shear load in the implanted related axis. The excessive translation of 3 mm was superimposed on lateral compensation, which corrects for the offset between the sample’s COR and the gimbal’s COR. The shear load, excessed translation, and the degree of rotation were determined at the subluxation point (**a**–**c**).

**Figure 9 bioengineering-11-00857-f009:**
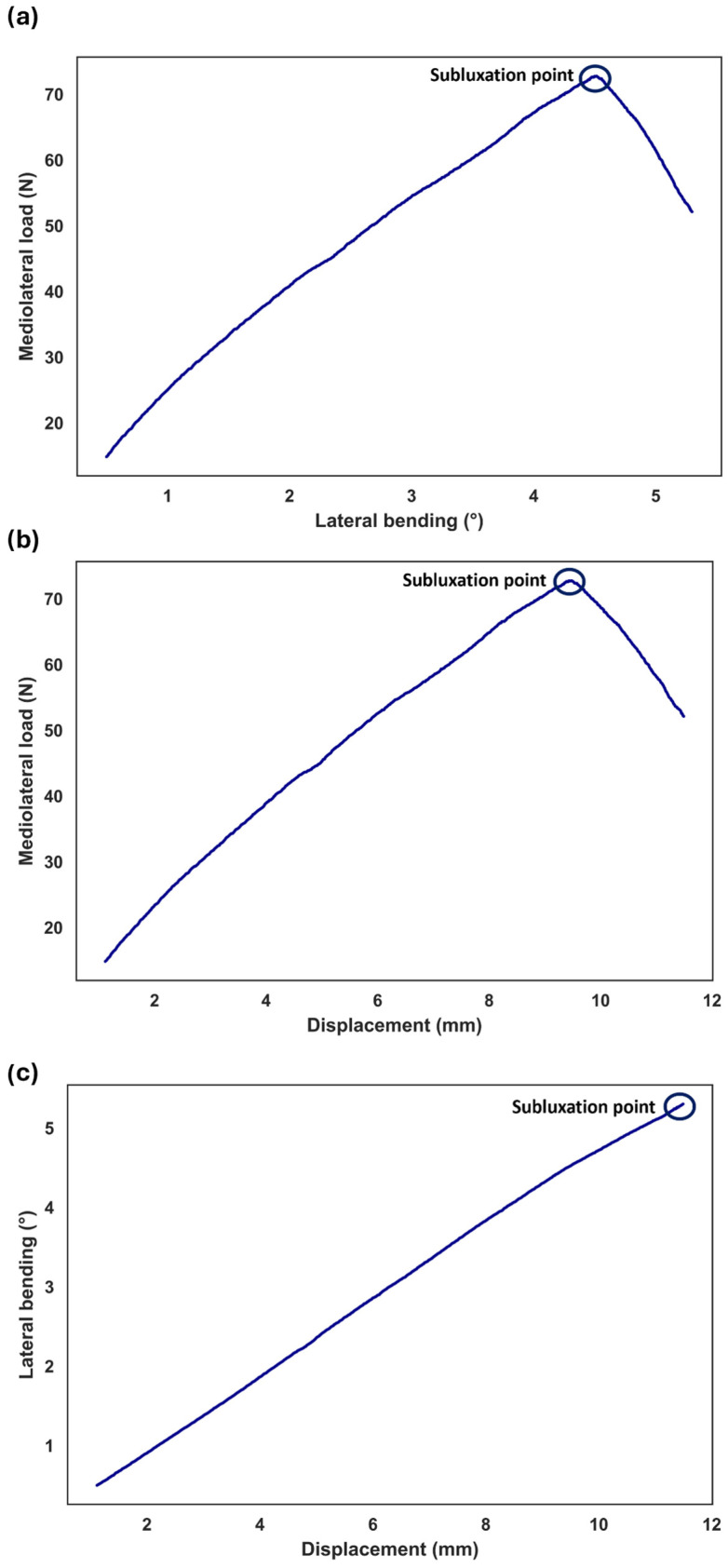
Subluxation test during lateral bending. Mediolateral load represents shear load in the implanted related axis. The excessive translation of 3 mm was superimposed on lateral compensation, which corrects for the offset between the sample’s COR and the gimbal’s COR. The shear load, excessed translation, and the degree of rotation were determined at the subluxation point (**a**–**c**).

**Table 1 bioengineering-11-00857-t001:** The reaction moments and forces for combined FE-LB, FE, and LB.

Motion Type	Tx (Std) Nm	Ty (Std) Nm	Tz (Std) Nm	Fx (Std) N	Fy (Std) N	Fz (Std) N
**Combined FE-LB**	0.56 (0.01)	0.82 (0.1)	0.01 (0.003)	20.92 (1.84)	13.40 (0.60)	106.61 (13.61)
**FE**	0.15 (0.06)	0.74 (0.12)	0.01 (0.01)	17.15 (2.14)	3.86 (1.09)	103.38 (29.94)
**LB**	0.67 (0.08)	0.1 (0.07)	0.01 (0.003)	2.57 (1.004)	14.87 (1.65)	102.46 (24.33)

**Table 2 bioengineering-11-00857-t002:** The resultant moment and force for combined FE-LB, FE, and LB.

Measurement Type	Combined FE-LB (Std)	FE (Std)	LB (Std)
**Resultant moment (Nm)**	1.0 (0.08)	0.75 (0.2)	0.68 (0.08)
**Resultant force (N)**	109.46 (13.26)	104.86 (29.51)	103.57 (24.07)

## Data Availability

The data are available within the article.

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
