# Peer review of "Evaluation of Load on Cervical Disc Prosthesis by Imposing Complex Motion: Multiplanar Motion and Combined Rotational–Translational Motion"

_bioengineering, 2024, doi:10.3390/bioengineering11080857_

Round 1

Reviewer 1 Report

Comments and Suggestions for Authors

“Five generic zirconia-toughened alumina (BIOLOX® delta, CeramTec, Germany) ball and socket samples (shape: cylindrical body, height: 20 mm, and diameter: 11 mm) were tested” Is this a relevant implant being used currently?

Figure 2, remove the wording “Command” from the legend.

Section 3.1, add a table to show the values, instead of two paragraphs being used.

Figure 8. Replace the images a) and b) by 3 projections of the curve in the 3 planes. It’s not possible to know the evolution of the curves only by the 3d curves shown.

Are the interfaces being continuously lubricated?

Reviewer 2 Report

Comments and Suggestions for Authors

Line 71-75 The research objective is too general. Give specific research questions, what you want to measure (force, rotation over time, etc.).

Line 241-243 "The first objective of this work was to compare the loads and moments in multiplanar motion tests ..." I did not notice such an objective in the introduction.

Line 266-267 "This study also investigated subluxation , which is one of the most important clinical concerns following disc replacement." Include this, for example, in the research question in the introduction.
